# Lower vaccine-acquired immunity in the elderly population following two-dose BNT162b2 vaccination is alleviated by a third vaccine dose

Laurent Renia[1,2,3] ✉, Yun Shan Goh[1,21], Angeline Rouers [1,21], Nina Le Bert [4,21], Wan Ni Chia [4,21], Jean-Marc Chavatte [5,6,21], Siew-Wai Fong[1], Zi Wei Chang[1,21], Nicole Ziyi Zhuo [7,21], Matthew Zirui Tay [1,21], Yi-Hao Chan[1], Chee Wah Tan[4], Nicholas Kim-Wah Yeo [1], Siti Naqiah Amrun[1], Yuling Huang[1], Joel Xu En Wong[1], Pei Xiang Hor[1], Chiew Yee Loh[1], Bei Wang [7], Eve Zi Xian Ngoh [7], Siti Nazihah Mohd Salleh [7], Guillaume Carissimo [1], Samanzer Dowla[7], Alicia Jieling Lim[5,6], Jinyan Zhang[4], Joey Ming Er Lim[4], Cheng-I Wang [7], Ying Ding[6], Surinder Pada[8], Louisa Jin Sun[9], Jyoti Somani[10], Eng Sing Lee [11], Desmond Luan Seng Ong[12], SCOPE Cohort Study Group*, Yee-Sin Leo [2,6,13,14], Paul A. MacAry[15,16], Raymond Tzer Pin Lin [5,6,15,22], Lin-Fa Wang [4,6,22], Ee Chee Ren[7,22], David C. Lye[2,6,14,17,22], Antonio Bertoletti[4,7,22], Barnaby Edward Young [2,6,14,22] & Lisa F. P. Ng [1,18,19,20,22]

Understanding the impact of age on vaccinations is essential for the design and delivery of vaccines against SARS-CoV-2. Here, we present findings from a comprehensive analysis of multiple compartments of the memory immune response in 312 individuals vaccinated with the BNT162b2 SARS-CoV-2 mRNA vaccine. Two vaccine doses induce high antibody and T cell responses in most individuals. However, antibody recognition of the Spike protein of the Delta and Omicron variants is less efficient than that of the ancestral Wuhan strain. Age-stratified analyses identify a group of low antibody responders where individuals ≥60 years are overrepresented. Waning of the antibody and cellular responses is observed in 30% of the vaccinees after 6 months. However, age does not influence the waning of these responses. Taken together, while individuals ≥60 years old take longer to acquire vaccine-induced immunity, they develop more sustained acquired immunity at 6 months post-vaccination. A third dose strongly boosts the low antibody responses in the older individuals against the ancestral Wuhan strain, Delta and Omicron variants.

---

A full list of affiliations appears at the end of the paper. *A list of authors and their affiliations appears at the end of the paper.
✉ e-mail: renia_laurent@idlabs.a-star.edu.sg

By the beginning of September 2021, over 250 million SARS-CoV-2 confirmed cases and five million associated deaths were reported worldwide (https://covid19.who.int/). https://jamanetwork.com/journals/jamanetworkopen/fullarticle/2781727?utm_source=silverchair&utm_medium=email&utm_campaign=article_alert-jamanetworkopen&utm_content=wklyforyou&utm_term=070921-zld210126r1 Several vaccines developed in record time have shown high efficacy against symptomatic infection and severe COVID-19. The Pfizer/BioNTech BNT162b2 mRNA vaccine, one of the most deployed worldwide, is a two-dose regimen, administered 21 days apart. Initial phase 3 data showed an efficacy of ~50 % after the first dose and >90% after the second dose against severe disease caused by the ancestral SARS-CoV-2 Wuhan strain in naïve individuals[1]. This was further supported by real-world vaccination data showing also high efficacy against[2–12]. Recent reports have shown that it still provides significant clinical protection against the emerging variants[13–18]. The BNT162b2 vaccine induces anti-Spike antibody, memory B cells and T cell responses in humans[19–25], which are both required for protection against infection and disease[26–35]; the former being considered as the main correlate of protection[35–39]. Different factors such as age, gender, microbiome, comorbidities influence the development of effective immune responses[40]. Since the elderly are at major risk of COVID-19 severe disease[41,42], it is necessary to assess their immune responsiveness to COVID-19 vaccination. Initial studies reported that antibody responses were similar to the younger groups[43], while others described lower responses in the older groups[43–48]. Despite lower immune responsiveness, the two-dose BNT162b2 vaccine has demonstrated similar vaccine efficacy in the elderly[49–53]. In the early days of vaccination implementation, there was discussion on delaying the second dose in order to offer a first dose to more individuals[54]. It was recently shown that delaying the second dose provided higher immunogenicity and maintained vaccine efficacy[55–57]. However, the kinetics of induction and maintenance of the adaptive immune responses in the elderly, which tend to respond less efficiently to vaccination[40], remain to be fully understood. A suboptimal immune response could favor breakthrough infections due to the ancestral or variant viruses[58–61].

Here, we compare the kinetics of specific antibodies, B and T cell memory responses in a cohort of BNT162b2-vaccinated healthcare workers and elderly individuals in Singapore up to 6 months post-immunization and for a subset of elderly low responders after a third dose. We specifically investigate longitudinal samples and integrate data from the same individuals with a variety of quantitative laboratory antibody, B and T cell assays, allowing a comprehensive analysis of the establishment and persistence of the vaccine-induced responses.

## Results

A cohort of 312 individuals was vaccinated with the Pfizer/BioNTech BNT162b2 vaccine from the beginning of January 2021–May 2021 in Singapore (Supplementary Table 1). The median age was 50.9 years (range, 22–82) and volunteers were predominantly female (58.3%) and Chinese (72.4%). Participants' characteristics differed across the different vaccination groups, which reflected vaccine prioritization for healthcare workers and elderly individuals. None of the participants had known or reported a history of SARS-CoV-2 infection and were all negative for antibodies against the N protein using the commercial Roche N serology assay. At the time of vaccination, Singapore had a low case count, which corroborated with low seroprevalence. To monitor immune responses, longitudinal blood samples were acquired at baseline corresponding to the day of the first dose, 21 days later at the time of the second dose, up to 180 days post first dose (Fig. 1a) and 1 month after a 3rd dose corresponding to a maximum of 300 days.

## Antibody response during and following two-dose SARS-CoV-2 mRNA vaccination

All volunteers ($n = 312$) were analyzed for vaccine-induced anti-Spike (S) protein-specific antibody levels and neutralizing efficacy using various assays. The flow cytometry-based assay (SFB) is based on the recognition of SARS-CoV-2 Spike protein stably expressed on the surface of HEK293T cells, allowing the detection of antibodies binding to different epitopes present on the full Spike protein[61,62]. The majority of volunteers seroconverted after the first dose (95% had higher antibodies than the cohort baseline and above their individual baseline) (Fig. 1b, Supplementary Table 2). After the second dose, all but one of the participants developed anti-Spike protein antibodies by day 90. Immunoglobulin isotyping showed that the proportion of vaccinees with detectable IgM (above both cohort and individual baseline) was >85% at day 21 but dropped to 12% by day 90 (12%) and was negligible by day 180 (Supplementary Fig. 1), indicating rapid maturation of the antibody after vaccination. Interestingly, IgG1 dominated the antibody response, followed by IgG3 and IgG2, while IgG4 was barely detected (Supplementary Fig. 1). By day 180, anti-Spike antibody levels had declined in 95% of the vaccinees (Supplementary Table 3) and on average by 39% (median binding percentage from 40.5% at day 90–24.1% at day 180). We also observed a sizeable proportion of low responders (individuals with responses below median cohort response at consecutive time points (37.2% at day 90 and 22.2% at day 180) (Supplementary Table 4)).

We next profiled antibodies specific to the receptor binding domain (RBD) of the S protein, which is the immunodominant target of anti-SARS-CoV-2 neutralizing antibodies[63] using a commercial assay (Roche S). The first vaccine dose induced antibodies in all but two vaccinees (Fig. 1c, Supplementary Table 2). After the second dose, all individuals seroconverted by day 90. However, 36.5% of individuals mounted a poor anti-RBD IgG response (Supplementary Table 5). A significant decline in anti-RBD antibody levels was also observed at day 180 in 77.8% of the vaccinees (Supplementary Table 3), on average by 30% (median value from 1140 U/ml at day 90 to 799.8 U/ml at day 180).

We next measured the level of neutralizing antibodies in these vaccinees using a surrogate virus neutralization test (sVNT) for the Wuhan strain, which has a good concordance with the live-virus neutralization test[64]. It was observed that >79.1% of the plasma had neutralizing antibodies above individual baseline after the first dose, 99% after the second dose and ~93% at day 180. However, between day 90 and day 180, serum neutralization efficacy declined in 77.5% of the participants and on average by 25% (from a median inhibition of 89.9–67.4%) (Fig. 1a, Supplementary Table 3). One-third of the vaccinees mounted a poor secondary neutralization response (Supplementary Table 6) and 19 out of 312 (6%) had no neutralizing antibodies (below baseline inhibition) at day 180 (Fig. 1d).

Notably, when the data from different serological assays were analyzed according to age, we observed a significant negative correlation of the age of the individuals with the antibody response at day 21 (after the first dose) and also with the antibody response at day 90 (after the second dose) (Supplementary Fig. 2). Sample distribution showed that low responders tended to cluster by age into two categories (1) <60 and (2) ≥60 years. Thus, the data were age-stratified and re-analyzed (<60 versus ≥60). For all assays, vaccinees ≥60 years had a lower response, compared with younger vaccinees after the first dose (Fig. 2b). After the second dose, the antibody responses were boosted in all groups. However, the increase was less pronounced in the older age group, who displayed lower antibody levels against the whole spike protein and RBD and had lower serum neutralizing capacity at day 90 (Fig. 2b). At day 180, while the elderly had more antibodies against the whole spike protein, the younger population (Fig. 2b, left panel) had lower levels of anti-RBD antibodies and lower neutralizing antibody capacity (Fig. 2b, middle and right panel, respectively). Similarly, the older individuals were among the low responders

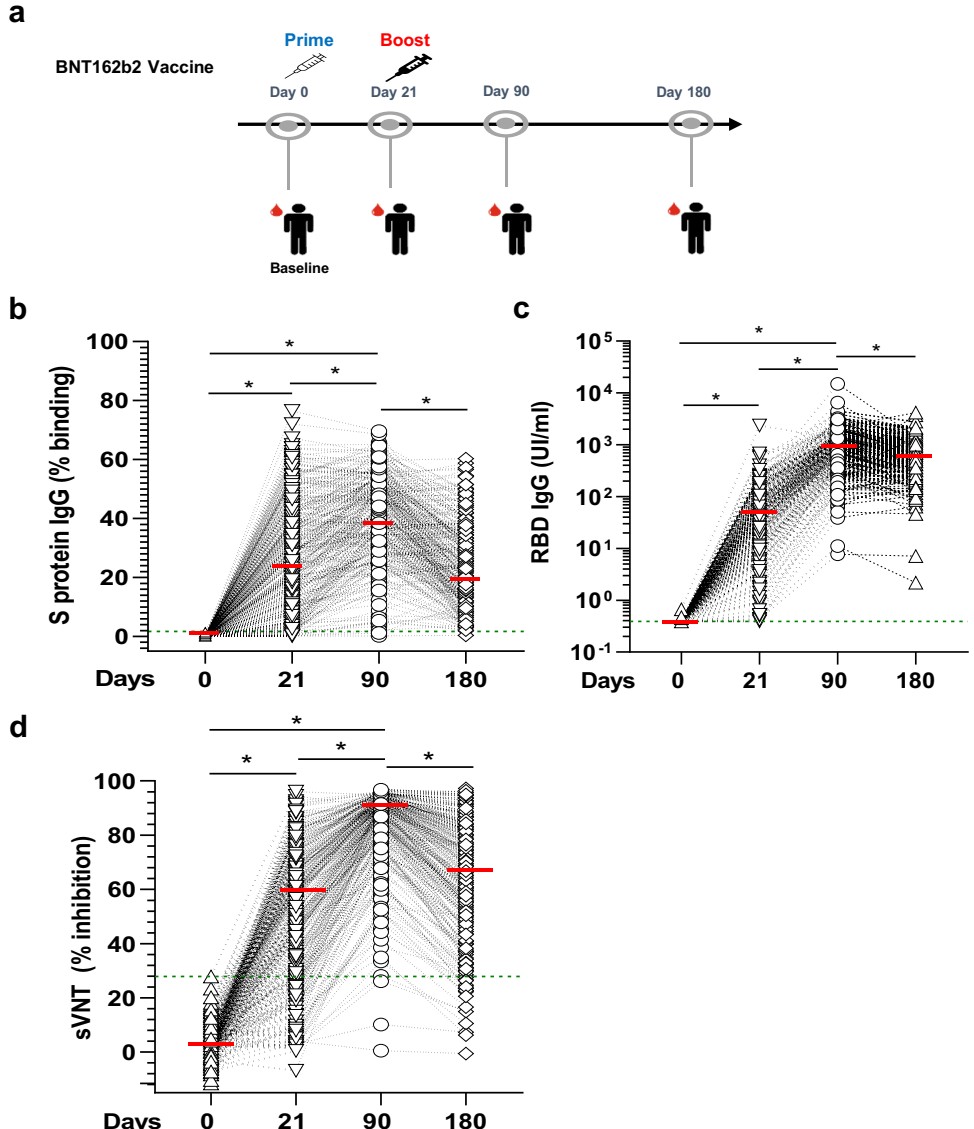

**Fig. 1 | Anti-SARS-CoV-2 spike protein antibody response after vaccination.**
**a** Schematic description of the longitudinal vaccination and blood sampling strategy in a cohort of Singaporean individuals ($n = 312$). Kinetics of IgG response were analyzed using three serological assays on paired samples. Overall cohort baseline value is defined as the value greater than maximum range of all samples in the cohort. The green dotted line represents the maximum range of the samples in the different assays. The median values of each group are represented by a red line. **b** A flow cytometry-based assay using the full Spike protein (SFB) assay. Median (range) of values at day 0 was 0.06% (0.002, 1.7). Antibody levels below the maximum range (1.7%) were considered baseline values. Median values was 28.4% at day

21, 42% at day 90 and 23.35%at day 180. *$p < 0.001$, Friedman test. **c** The Roche S assay using the RBD protein fragment. Median (range) of values at day 0 was 0.39 U/ml (0.39, 0.67). Antibody levels below the maximum range (1.38 U/ml) were considered baseline values. Median values were 40.55 U/ml at day 21, 806.2 U/ml at day 90 and 603 U/ml at day 180. *$p < 0.001$, Friedman test. **d** A surrogate virus neutralization test (sVNT). Median (range) of values at day 0 was 0.39% (−11.54, 27.94). Inhibition below the maximum range (27.94%) were considered baseline values. Median values were 56.4% at day 21, 89.9% at day 90 and 67.4% at day 180 *$p < 0.0001$, Friedman test.

(participants with responses below median cohort response at consecutive time points) (Supplementary Tables 4, 5 and 5).

We next assessed the waning of antibodies between age groups by measuring the difference in antibody levels between days 180 and 90 in paired samples (Fig. 2c). Although the antibody levels were lower at the cohort level, the decline in antibody levels was significantly more pronounced in the older population than in the younger one (Fig. 2c left and middle panels, Supplementary Table 3). The waning of neutralization capacity did not differ between both age groups (Fig. 2c right panel, Supplementary Table 3).

We also examined the binding efficiency of the vaccinated plasma to the spike protein of the Delta (B.1.617.2) or the Omicron (B.1.1.529, BA1 substrain) variants using the SFB assay. In a previous

study, we reported that IgG levels against the Wuhan ancestral strain or its variants were strongly correlated with their capacity to inhibit pseudovirus and live-virus neutralization expressing the respective various Spike proteins[65,66]. Here, we show that at any time points, the antibody response against the Delta variant was lower than the Wildtype ancestral strain and extremely low or non-existent against the Omicron variant (Fig. 3a). However, at day 21 (after the first dose), the difference in recognition was only significant between the two age groups for the Wuhan ancestral strain and Delta but not the Omicron variant, and, at day 90, for the Wuhan strain (Fig. 3a, left and middle panels). This was not observed at day 180, where the difference in recognition was similar in both age groups (Fig. 3b, middle and right panels).

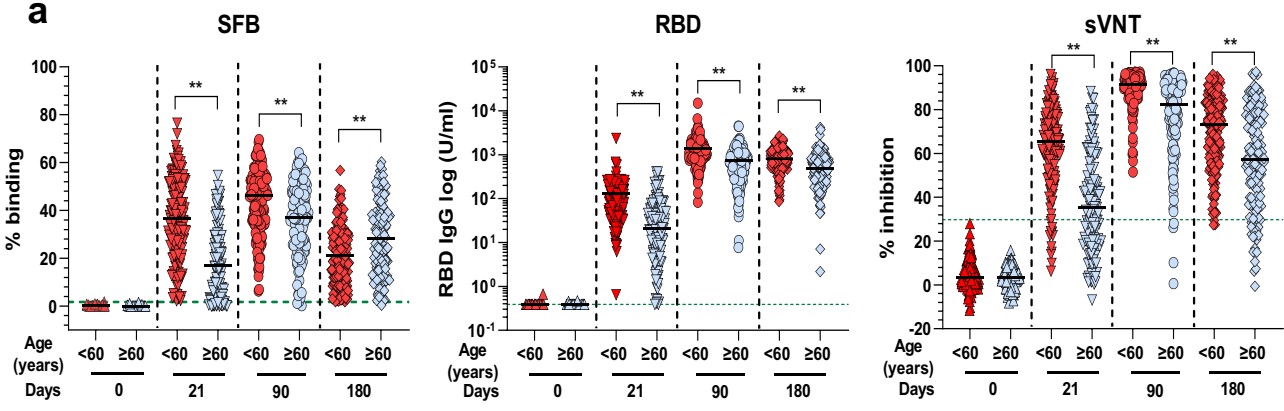

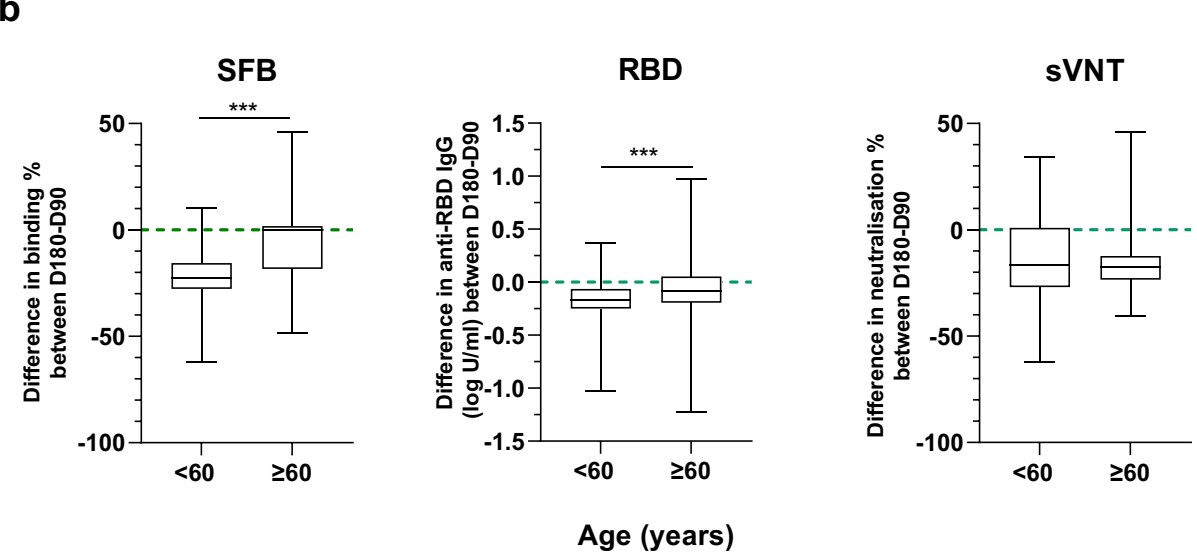

**Fig. 2 | Age stratification of antibody responses. a** Comparison between age groups: <60 (n = 178) and ≥60 (n = 134), at different time points (day 0, 21, 90 and 180) using SFB (left panel); RBD Roche S assay (middle panel); and sVNT assay (right panel). The median value of each age group is represented by the black line. The green dotted lines represent the maximum range of the samples for the whole cohort baseline as defined in Fig. 1. *p < 0.01, two-sided Mann–Whitney test. **b** Box plots showing difference in antibody response between days 180 and 90 for paired samples between the different age groups (<60 (n = 178) and ≥60 (n = 134)). Data are represented as median (middle line) with 25th, 75th percentile (box) and 5th and 95th (whiskers). Median values are −22.5 for <60 and 0.02 for ≥60 for the SFB assay; −358 for <60 and −72 for ≥60 for the Roche S assay; and −16.75 for <60 and −17.3 for ≥60 for the sVNT assay. ***p < 0.0001, two-sided Mann–Whitney test.

## Memory B cell response during and following two-dose SARS-CoV-2 mRNA vaccination

To measure the vaccine-induced RBD-specific circulating memory B cells, B cell ELISPOT assay[67,68] was performed on a subset of randomly selected age-matched individuals (n = 78, from which we had 36 paired samples for the four time points). There was no significant increase after the first dose at day 21, even though 47% of the individuals with paired samples had higher memory B cells than their baseline (Supplementary Table 2). After the second dose, a significant increase in the percentage of RBD-specific memory B cells was observed at day 90 (Fig. 4a). Analysis of paired samples confirmed these observations (Fig. 4b), where 76.5% had positive responses above their baseline levels. By day 180, the numbers of RBD-specific B cells continued to increase (Fig. 4a), with 85.3% of individuals having responses above their baseline levels at day 180 (Supplementary Table 2). Generally, all individuals had produced RBD-specific circulating B cells in 6 months.

When the data were age-stratified, we observed that the specific memory B cell response was lower in vaccinees ≥60 years after the first dose at day 21 than vaccinees <60 years (Fig. 4c). However, after the second dose, the specific memory B cell response increased in vaccinees ≥60 years at day 90 and 180 (Fig. 4c), corresponding to an overall increase in the number of total memory B cells (Fig. 4d). After two doses, the specific memory B cell response continued to increase for both age groups over time (Fig. 4c). At day 180, the difference between the two age groups disappeared, with both age groups having similar levels of memory B cell response (Fig. 4c, d). By comparing differences in the memory B cell response between time points (Fig. 4e), we found that younger individuals responded faster, with a greater increase right after the first dose at day 21 (p < 0.001, Mann–Whitney test). In contrast, the older age group had a substantial increase at day 90, which was higher than the younger age group (p < 0.001, Mann–Whitney test), demonstrating the importance of the second dose for the older age group (Fig. 4e).

## T cell responses during and following two-dose SARS-CoV-2 mRNA vaccination

T cell stimulation was determined by quantifying cytokines (IL-2 and IFN-γ) directly secreted by Spike-specific CD4 and CD8 T cells in whole

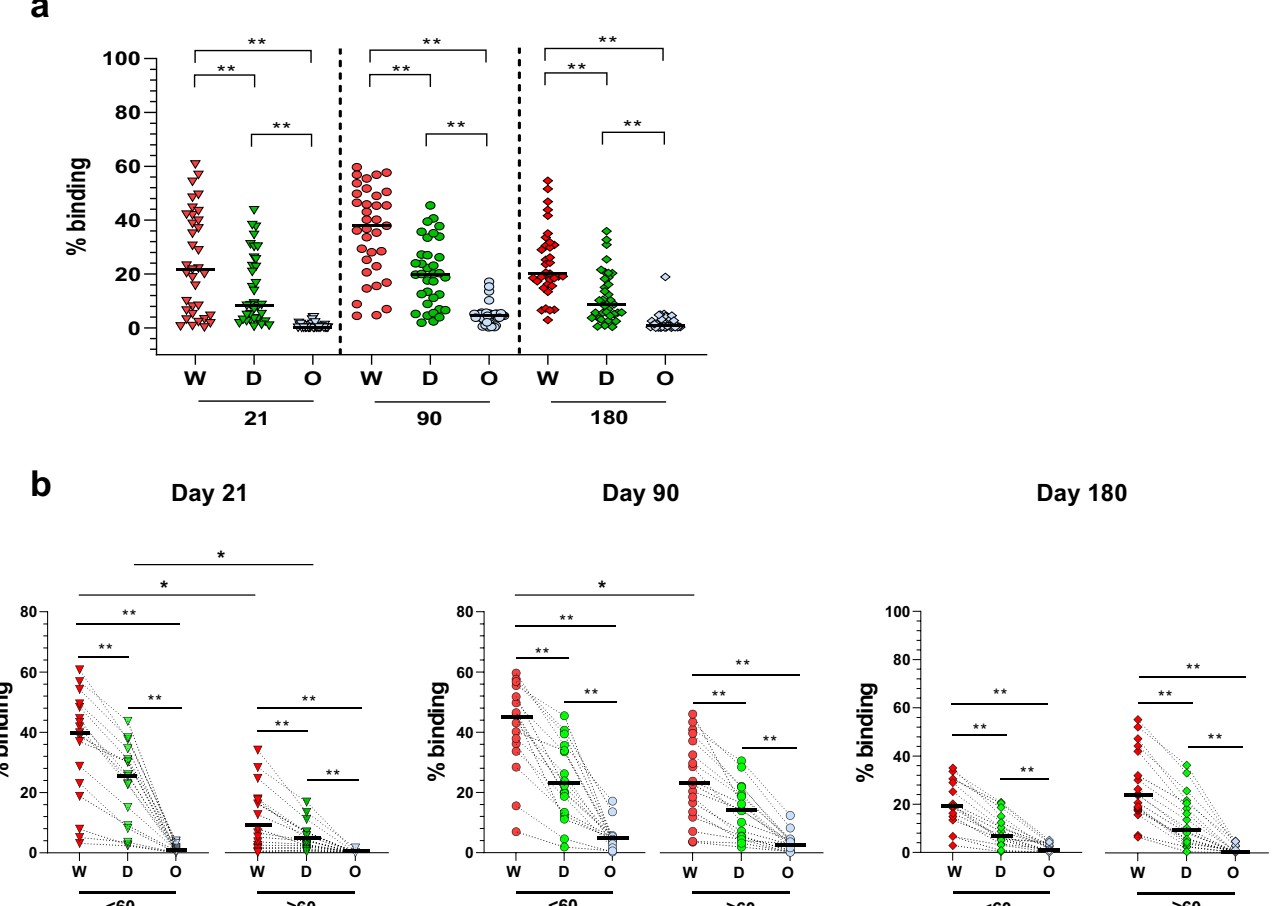

**Fig. 3 | Antibody recognition of Wuhan ancestral strain, Delta and Omicron variants. a** Comparison of antibody response of the vaccinees ($n = 35$) at days 21, 90 and 180 using the SFB assays with cells expressing either the Wuhan ancestral virus (W), the delta (D) or Omicron (O) variants of the Spike protein. Median values of the samples were at day 21, 90 and 180 respectively 39.8%, 37.9% and 20.2 for the Wuhan strain; 8.45%, 20% and 8.8% for Delta;0.13%, 4.5% and 1.1% for Omicron. **$p < 0.001$, two-sided Mann–Whitney test. **b** Comparison between age groups: <60 ($n = 17$) and ≥60 ($n = 18$), at different time points (days 21, 90 and 180) using SFB with cells expressing either the wild-type Wuhan ancestral virus (W) or the delta (D) or Omicron (O) variants of the Spike protein. The median value of each group is represented by a red line. They were respectively at days 21, 90 and 180: Wuhan ancestral strain, 39,8%, 45,4% and 19.1% (<60) and 9%, 28.7%, and 23.96% (≥60); Delta variant: 25.4%, 22.2% and 7% (<60) and 5%, 17.4%, and 9.5% (≥60); and Omicron variant: 0.8%, 5% and 7% (<60) and 0.06%, 3.8%, and 0.4% (≥60). **$p < 0.001$, Friedman test when the three strains were compared together, or two-sided Mann–Whitney test when the same strain was compared between the two age groups.

blood, following overnight incubation with peptide pools covering 75–80% of Spike protein[69]. This was done in a subset of volunteers ($n = 160$) randomly selected from the cohort but age-matched ($n = 82 < 60$ and $n = 78 ≥ 60$). At baseline, majority (~95%) of the individuals had very low production of IL-2 or IFN-γ (<10 pg/ml) after Spike-peptide pool stimulation (Fig. 5a and b). After one dose, most of the vaccinees had a T cell response that increased further after the second dose. 98.7% of the vaccinees had a peptide-mediated IL-2 response above individual baseline after the first dose at day 21, days 90 and 180 (Fig. 5a, Supplementary Table 2). A robust IFN-γ response above baseline was also observed (~93 to 95% after the first and second doses) and sustained up to day 180 (Fig. 5b, Supplementary Table 2).

We next performed a detailed analysis of the T cell subsets by ELISPOT in a smaller subset of the volunteers due to cell availability. We used peptides covering potential CD8 or CD4 T epitopes (see materials and methods). For the CD8 assay using Spike protein peptide pools covering potential 9mers CD8 epitopes[69,70], we showed that, at baseline, the CD8 T cell response was already high in some vaccinees (Fig. 5c), suggesting a cross-reactive CD8 T cell response from exposure to other circulating coronaviruses. After the first dose, 54% had an increase in spots above their individual baseline values at day 21. After

the second dose, 75% of the vaccinees had a response above their individual baseline at day 90 (Fig. 5c, Supplementary Table 2). By day 180, only 40.3% still had a CD8 T cell response (above their own baseline values, Supplementary Table 2). Overall, 88.9% (64/72) mounted a CD8 T cell response during the 6-month follow-up. However, a comparison between responses at day 180 and 90 showed that the response waned in 48% of the vaccinees (Supplementary Table 3).

We next stimulated PBMC with a 15 mer peptide pool corresponding to potential CD4 epitopes[22] and measured the response by ELISPOT. CD4 Th1 (IL-2 and/or IFN-γ) responses were low at baseline, except for a few individuals (Fig. 6d). After one dose, 69% of the vaccinees and 84.6% after the second dose had a response higher than their baseline by day 90 and 83.33% by day 180 (Supplementary Table 2). Overall, the CD4 Th1 cell response was significantly different after the first dose and further significantly boosted after two doses. Over 96.2% (75/78) mounted a response during the 6-month follow-up. Comparison between responses at day 90 and 180 showed that the CD4 Th1 response waned in 46.6% of the vaccinees (Supplementary Table 3). A CD4 Th2 cell response was observed but was not as strong as the Th1 response (Supplementary Fig. 3). At day 21, 59% of the vaccinees had values above their own baseline, a percentage which

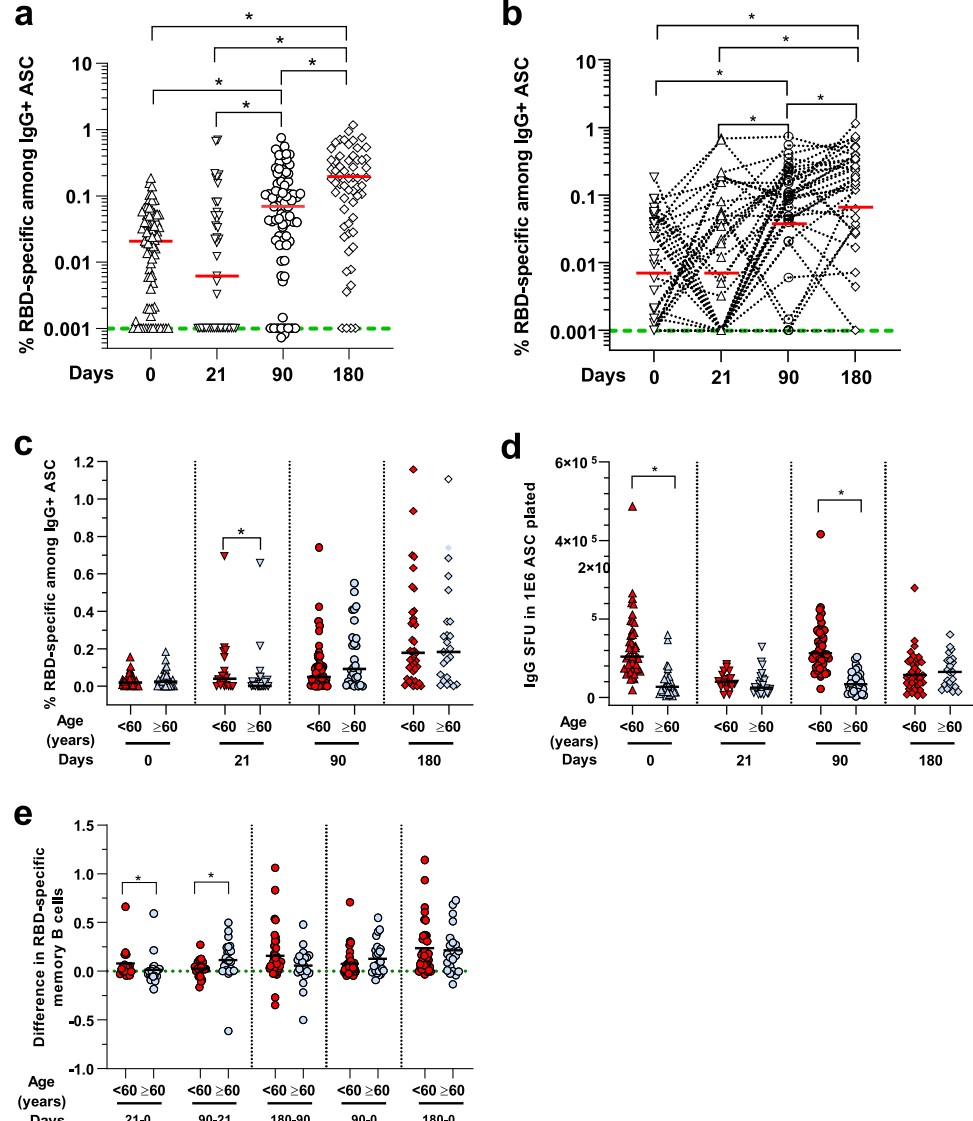

**Fig. 4 | Circulating RBD-specific memory B cells after vaccination. a** RBD-specific memory B cells were determined by ELISPOT using a RBD protein. Determination of % RBD-specific memory B cells among IgG+ antibody-secreting cells (ASC) done on PBMC from vaccinated participants at baseline or day 0 (n = 73), at day 21 (n = 43), at day 90 (n = 76) and at day 180 (n = 60). The median value of each group is represented by a red line and were at day 0, 21, 90 and 180, respectively 0,02%, 0.006%, 0.07% and 0.18% of the % RBD-specific memory B cells among IgG+ ASC. *p < 0.01, Dunn's test after Kruskal–Wallis (p < 0.001) on log-transformed data. Green dotted lines indicate the limit of detection for the assay. **b** Paired wise comparison of % RBD-specific memory B cells among IgG+ ASC for the analyzed aged group at different days post doses is shown (n = 35). The median value of each group is represented by a red line and is the same as in (**a**). *p < 0.01, Friedman test on log transformed data. **c** RBD-specific memory B cells comparison between the analyzed age groups. Samples were from individuals: aged <60: at day 0 (n = 46), day 21 (n = 18), day 90 (n = 46), and day 180 (n = 37); and aged ≥60, at day 0 (n = 27), at day 21 (n = 25), at day 90 (n = 30), and day 180 (n = 23). **d** Total IgG producing memory B cells comparison between the same analyzed age groups as above in (**c**). The median value of each group is represented by a black line. *p < 0.01, two-sided Mann–Whitney test. **e** Difference in RBD-specific memory B cells between paired samples and different time points (n = 35). *p < 0.01, two-sided Mann–Whitney test.

remained constant at day 90 but started to wane by day 180 (Supplemental Tables 2 and 3).

Age-stratified analysis showed that post-vaccination Spike peptide pool-mediated IL-2 response was similar in both age groups at all time points (Fig. 6a). The IFN-γ response was lower at baseline in the <60 group. However, after the first dose (day 21), it reached a similar level to that of ≥60 group. At days 90 and 180, the older age group had T cells producing significantly more IFN-γ than the younger individuals (Fig. 6b). We did not observe any age effect on the CD8 ELISpot response (Fig.6c). CD4 Th1 was significantly lower at baseline for the older age group but the responses were similar at days 21, 90 and 180 (Fig. 5d). Post-immunization Th2 cell responses were also similar at the different times (Supplementary Fig. 3, Supplementary Table 3).

We next assessed the waning of T cell responses between age groups by measuring the difference in response levels between days 180 and 90 in paired samples (Fig. 6e, Supplementary Table 3). Although T cell responses were lower at the cohort level, the decline was not significantly different between age groups. On the contrary, IFN-γ T cell response was even higher in the older age group (Fig. 6e, middle left panel).

**Immune responses in individuals ≥60 years following a booster vaccination with a third dose of the SARS-CoV-2 mRNA vaccine**
Our data above and others[45–48] indicated that a significant fraction of older adults initially mounted a lower response to the two-dose vaccination. Thus, in September 2021, individuals ≥60 were identified as a

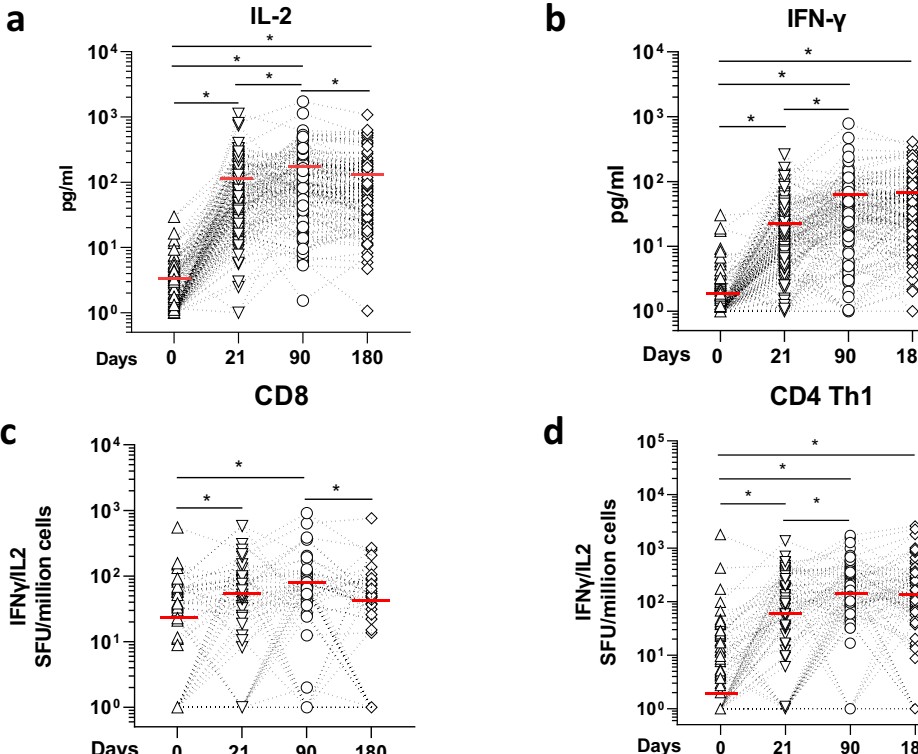

**Fig. 5 | Anti-SARS-CoV-2 spike protein T cell responses. a** IL-2 and **b** IFN-γ secretion profile of whole-blood cultures stimulated with S protein peptide pool compared at different time points of paired samples from vaccinated individuals ($n = 160$). The limit of detection for each cytokine (IL-2 = 5.4 pg/ml; IFN-γ = 1.7 pg/ml). Values below limit of detection levels were plotted as 1. *$p < .001$, ANOVA on log-transformed data, which follow a normal distribution. The mean values of each group are represented by a red line and were at day 0, 21, 90 and 180: 1.8 pg/ml, 23.8 pg/ml, 55.4 pg/m and 57.2 pg/ml for IL-2; 2.96 pg/ml, 102.6 pg/ml, 154.2 pg/ml and 123.9 pg/ml for IFN-γ, respectively. Kinetics of Spike-protein-specific CD8 **c** or

**d** CD4 Th1 cells overtime in vaccinees. T cells were assayed on a subset of vaccinees ($n = 80$) by IL-2/ IFN-γ ELISPOT using 9 mer or 15 mer pool peptides, respectively. Data are presented are spot forming units (SFU) per million of PBMC from paired samples from vaccinated individuals at four time points. Each data point represents the normalized mean spot count from duplicate wells for one study participant, after subtraction of the medium-only control. The median values of each group are represented by a red line and were at day 0, 21, 90 and 180: 26.4, 46.2, 55.4 and 57.2 SFU for CD8 T cells; and 2, 67.25, 167.4 and 134.5 SFU for CD4 Th1 cells *$p < 0.01$, Dunn's test after Kruskal–Wallis ($p < 0.001$) on log-transformed data.

priority population and were recommended for a booster vaccination (a third dose of BNT162b2 or mRNA 1273 [Spikevax, Moderna] vaccine) in Singapore. Here, we analyzed the effect of the booster vaccination in a subset of older individuals from our cohort, who received their booster BNT162b2 vaccination between day 189 to 270 after the first injection (Fig. 7a). Blood samples were taken ~30 days after booster injection. We observed that the third dose strongly boosted the antibody responses against the total Spike protein or its RBD (Fig. 7b and c). The boosting injection also induced a strong antibody response against the spike protein of the Delta and Omicron variants (Fig. 7b). When the samples were analyzed in the wildtype surrogate virus neutralization assay, inhibition was boosted to high levels (>80%) following booster vaccination (Fig. 7d). This was also true in a pseudo-virus neutralization assay using Wuhan, Delta, and Omicron pseudoviruses (Fig. 7e). In line with the antibody responses, memory B cell response was also strongly boosted in all individuals (Fig. 7f). When the T cell responses were analyzed, we found that the IL-2 T cell response was lower but still high (Fig. 7g). The IFN-γ T cell response remained unchanged (Fig. 7h), matching the levels of CD4 Th 1 cells detected by ELISPOT (Fig. 7i). The CD8 T cells were boosted by the 3rd dose (Fig. 7J). Of note, ~10% of the vaccinees mounted poor T cell responses even after the third dose, despite mounting good antibody responses (Fig. 7g to i).

## Discussion
In this study, we show that the two-dose regimen Pfizer/ BioNTech BNT162b2 COVID-19 vaccine is highly immunogenic and generates

robust antibody, B and T cell responses against the Spike protein of the ancestral Wuhan strain in most Singaporean vaccinees (>75%). Despite the strong immunogenicity, a sizeable proportion of the vaccinees mounted a low antibody response. Further analysis showed that individuals ≥60 years developed antibody responses at a slower pace, with a lower peak, and were more represented in the low responders' fraction (Fig. 2, Supplementary Tables 3–5). However, the antibody responses decreased less rapidly in the older age group as seen 6 months post-immunization (Fig. 2b, middle and right panel, Fig. 2c). These data are in line with the memory B cell data, with lower levels of memory B cell in the older age group after the first dose at day 21, but eventually were at similar levels at day 90 and continued to increase at day 180. This indicates the building of equivalently strong B cell memory in responding older individuals as in younger individuals. These findings agree with recent studies showing that recall of RBD-specific memory B cells is stable following the 2 dose-vaccination regimen 6–9 months after the first injection[71–75]. The over-representation of older individuals in the low responder groups is likely a consequence of immunosenescence, which is characterized by the reduced adaptive immune responses[76–79]. This has been well described for influenza vaccines in Caucasian populations[80]. However, studies on the Chinese population in Singapore showed no impact of age on immunogenicity after influenza vaccination[81]. Here, we showed that, following COVID-19 vaccination, antibody responses were partially affected by age. Additional genetic, behavioral, nutritional or environmental factors might account for this phenomenon and deserve further studies. The low neutralizing antibody response in a

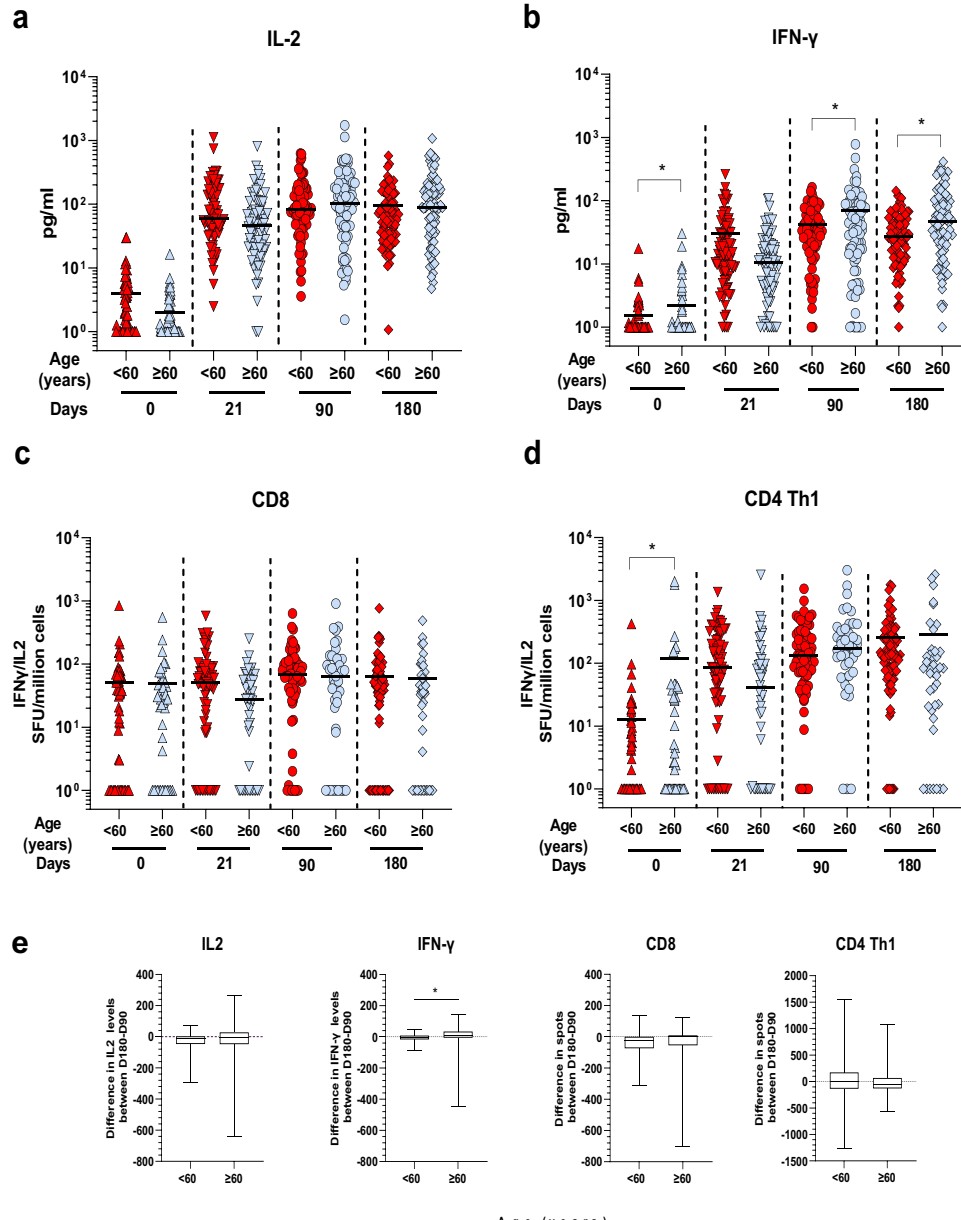

**Fig. 6 | Age stratification of T cell responses.** Comparison of the T cell response between age groups of samples from vaccinated individuals at different time points post immunization. **a** IL-2 and **b** IFN-γ production induced by Spike-peptide pool stimulation from individuals aged <60: (n = 82), and ≥60 (n = 75). Mean values are indicated by a dark line. *p < 0.01, two-sided Student *t*-test on normalized log values. **c** CD8 T cells comparison between the analyzed age groups. Less than 60 group: day 0 (n = 66), day 21 (n = 66), day 90 (n = 66) and day 180 (n = 85), and ≥60: day 0 (n = 43), day 21 (n = 44) and day 90 (n = 43) and day 180 (n = 41). Median values are indicated by a dark line. *p < 0.01, two-sided Mann–Whitney test on log values. **d** CD4 Th1 cells comparison between the analyzed age groups. Less than 60 group: day 0 (n = 72), day 21 (n = 72), day 90 (n = 72) and day 180 (n = 83), and ≥60:

day 0 (n = 43), day 21 (n = 44) and day 90 (n = 43 and day 180 (n = 41). Median values are indicated by a dark line. *p < 0.01, two-sided Mann–Whitney test. *p < 0.01, Mann–Whitney test on log values. **e** Box plots showing difference in T cell responses measure in the different assays between days 180 and 90 for paired samples for both age groups [IL-2 and IFN-γ: <60: (n = 82), and ≥60 (n = 75); CD8 T cells: <60: (n = 45), and ≥60 (n = 27)) and CD4 T cells, <60: (n = 51), and ≥60 (n = 28)]. Data are represented as median (middle line) with 25th, 75th percentile (box) and 5th and 95th (whiskers). Median values of difference levels are: −12.8 for <60 and −4 for ≥60 for IL-2; −5.1 for <60 and 10.3 for ≥60 for IFN-γ; −73.8 for <60 and −54.5 for ≥60 for CD8 T cells; and 10.5 for <60 and −53.2 for ≥60 for CD4 Th1 cells; *p = 0.003, two-sided Mann–Whitney test.

larger subset of the older age group (20–30% more than in the younger groups) has important clinical implications. High neutralizing antibody levels have been proposed as one of the essential protective mechanisms against infection with the ancestral Wuhan strains of SARS-CoV-2. They are also required to protect against, albeit less efficiently, new emerging variants such as Delta or Omicron that can escape antibody neutralization[65,66,82–98]. Our study showed that the vaccine-induced antibody reactivity against the Delta variant was lower, and extremely low or non-existent against the Omicron variant,

compared with the Wuhan ancestral strain. This lower response against the variants was more pronounced for Delta after the first injection in older individuals. However, at a later time-point, the responses against both variants were similar across the different age groups, and the waning was not limited to the older individuals.

Spike-specific T cell responses were induced in most vaccinees (>95%) and remained high until day 180 (Fig. 4). Unlike the antibody response, both age groups were equally represented in the low T cell responder group (Supplementary Tables 7–9). Although around

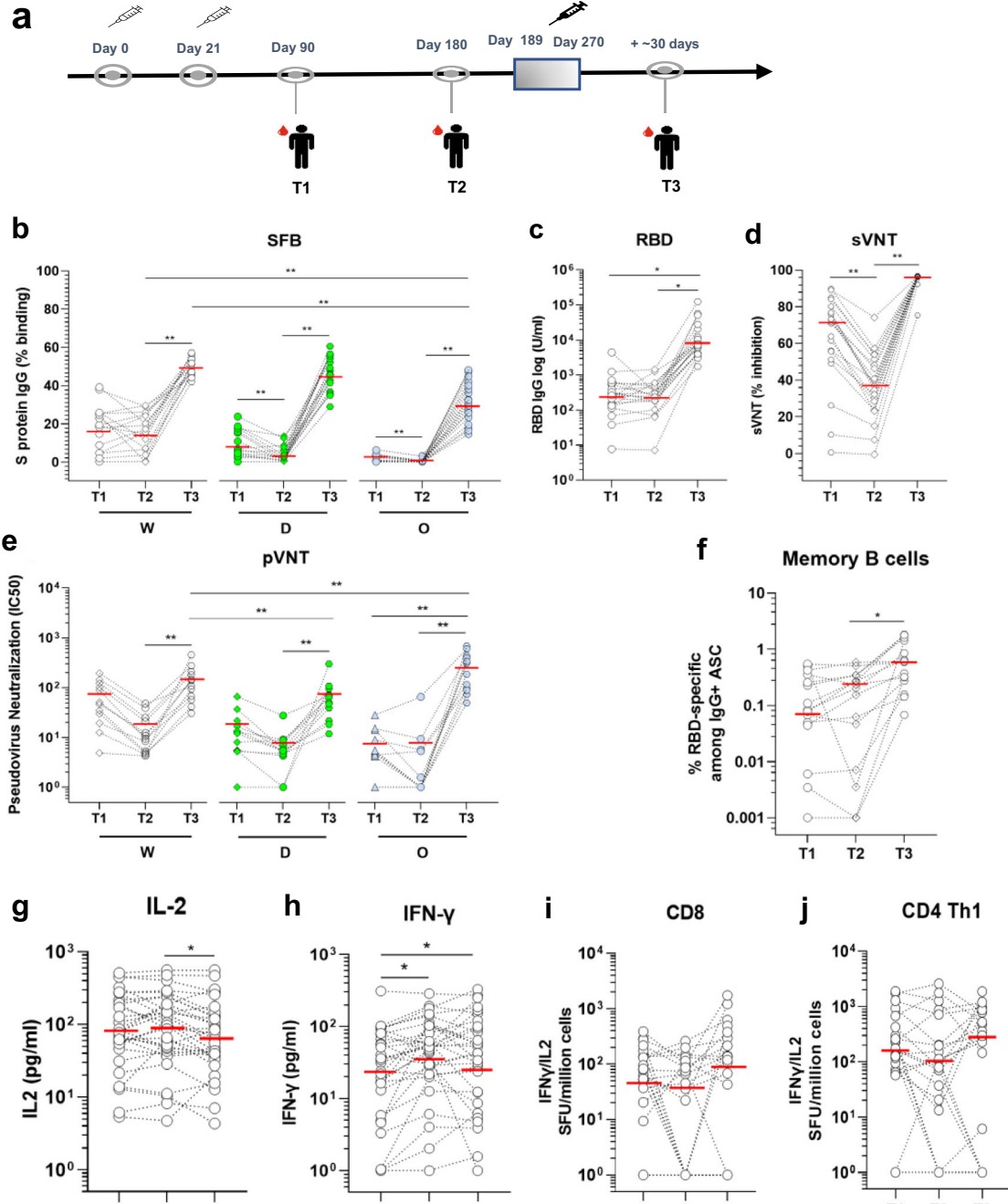

**Fig. 7 | Anti-SARS-CoV-2 spike protein antibody and T cell response before and after the third booster dose. a** Blood sampling strategy in subsets of the cohort of Singaporean individuals mentioned in Fig. 1. Kinetics of IgG response were analyzed using three serological assays on paired samples taken at time T1 (day 90), T2 (day 180), and T3 (day 189–270) post first injection. **b** SFB assay using the cells expressing the Wuhan ancestral strain (W, white dots), Delta (D, green dots) or Omicron variant (O, blue dots). Median of group values at T1 ($n = 16$), T2 ($n = 20$), and T3 ($n = 20$) were: 19, 16.45, and 50.2% (W); 6.7, 2.56, and 45,73% (D); and 1, 0.18, and 31.65% (O). $p < 0.001$, Friedman test; **c** Anti-RBD antibodies using the Roche S assay. Median of values ($n = 20$) at T1, T2, and T3 were 277, 246.8 and 7723 U/ml; *$p < 0.001$, Friedman test. **d** Surrogate virus neutralization test (sVNT). Median values ($n = 20$) at T1, T2, and T3 were 71.2, 36,94 and 96.4% of inhibition. *$p < 0.0001$, Friedman test. **e** Neutralization assay using pseudoviruses expressing the SARS-CoV-2 Spike of the Wuhan ancestral strain (W), Delta variant (D) or the Omicron variant (O) ($n = 12$). Median of IC50 values at T1, T2, and T3 are 48.9, 10.2, and 123.2 (W); 12.5, 5.3, and 57.1 (Delta); 5.0, 1.0, and 248.9 (Omicron) respectively. *$p < 0.01$, Friedman test. **f** RBD-specific memory B cells. Paired wise comparison of total RBD-specific memory B cells for the analyzed aged group at different days is

shown ($n = 15$). Median of values at T1, T2, and T3 are 0.08%, 0.2 and 0.6 % of total PBMC. **$p < 0.001$, Friedman test. **g** IL-2 and **h** IFN-γ secretion profile of whole-blood cultures stimulated with S protein peptide pool compared at the three time points of paired samples from vaccinated individuals ($n = 31$). The limit of detection for each cytokine (IL-2 = 5.4 pg/ml; IFN-γ = 1.7 pg/ml). Values below limit of detection levels are plotted as 1. Median of values at T1, T2, and T3 are: 81.2, 96.6 and 62.8 pg/ml for IL-2, and 31.2, 52 and 43 pg/ml for IFN-γ and are indicated as red lines. *$p < 0.01$, two-way ANOVA on log-transformed data, which follow a normal distribution. Kinetics of Spike-protein-specific CD8 (**i**) or CD4 Th1 cells (**j**) over time in vaccinees. T cells were assayed on a subset of vaccinees ($n = 11$) by IL-2/ IFN-γ ELISPOT as in Fig. 5. Data are presented are spot forming units (SFU) per million of PBMC from paired samples from vaccinated individuals at three time points. Each data point represents the normalized mean spot count from duplicate wells for one study participant, after subtraction of the medium-only control. Values below limit of detection levels are plotted as 1. Median of values at T1, T2, and T3 are: 79, 45.3 and 118.9 SFU for CD8 T cells, and 173, 105.1 and 295.9 SFU for CD4 Th1 cells and are indicated as red lines. *$p < 0.01$, Dunn's test after Kruskal–Wallis test.

30–50%, depending on the assay used, of the vaccinees, experienced a small (<20%) decrease in response between days 90 and 180 (except for the IFN-γ T cell responses), waning was not affected by age. This is particularly important as T cells are thought to protect against severe disease[28–34]. They recognize peptide epitopes distributed throughout the SARS-CoV-2 Spike protein and in other viral proteins[68,70,99–101] and are less susceptible to antibody escape mutations in variant strains[102–109]. Our findings showed that the levels and activities of the T cell response were maintained in both age groups up to 180 days, suggesting protection in all age groups. This agrees with recent studies, which have reported strong efficacy of the BNT162b2 vaccine against severe disease after infection with the Delta (80-95%) or Omicron (70-80%) variants, compared with the Wuhan ancestral strain[13–18,110,111].

Our findings show that although the antibody responses have started to wane, recall T cell responses remain stable. This confirmed recent findings obtained in vaccinees in different vaccinated populations[25,112–115]. It is widely accepted that high antibody levels are essential for protection against infection and T cells against severe disease[27,28]. This is supported by real-world data that have demonstrated a gradual decline in or limited vaccine efficacy against infection with the ancestral strain and with Delta or Omicron, respectively, but sustained high protection against hospitalization and death up to 6 months after the second dose[9,116–118].

To prevent waning or increase immune responses in poor responders, the Singapore health authorities opted for an additional booster vaccine doses with the same mRNA vaccine platforms. We demonstrated here that a booster vaccination in our elderly low responders significantly increased both antibody and T cells responses against the Wuhan strain, the Delta and Omicron variants (Fig. 7). Many countries have also initiated additional booster vaccine doses with same or different platforms, showing similar effects of the booster doses in terms of improved antibody and T cell responses against the Wuhan strains and variants[109–111,119–130], protection against infection and severe disease in individuals across all age groups[131–137].

Lastly, the longevity of the immune response after a third dose is unknown. However, waning antibody levels and protection against infection has been recently reported in patients who have received three doses of the BNT162b2 vaccine[138,139]. In addition, we have also shown in this study that a small fraction of the vaccinees (~10%) (depending on the assays) mounted poor T cell responses even after the third dose, despite mounting good antibody responses. It has also been shown that cancer patients or individuals under immunosuppressive drug treatments also mount poor vaccine response even after a third dose[140]. Countries, like Israel, have implemented a 4th dose for the low responders. However, although the fourth vaccination raises antibody levels, the increase in protection against SARS-CoV-2 infection was modest[141]. Short-term repeated vaccinations may not be logistically feasible and may induce vaccine fatigue. Thus, second generation vaccines with new platforms, better immunogens or adjuvants[142] that induce a more rapid and efficient helper T cell and potent CD8 T responses are needed[143].

## Methods
### Cohorts and ethics
A cohort of 312 individuals was recruited (Supplementary Table 1) comprising healthcare workers and older individuals. Our study complies with all the relevant ethical regulations. The study design and protocol for the COVID-19 PROTECT study group were assessed by National Healthcare Group (NHG) Domain Specific Review Board (DSRB) and approved under study number 2012/00917. Collection of healthy donor samples was approved by SingHealth Centralized Institutional Review Board (CIRB) under study number 2017/2806 and NUS IRB 04-140. Written informed consent was obtained from all study participants in accordance with the Declaration of Helsinki for Human

Research. The experiments adhered to the principles set out in the Department of Health and Human Services Belmont Report.

### Sample collection
Blood was collected in VACUETTE EDTA tubes (Greiner Bio, #455036) or in Cell Preparation Tubes (CPT) (BD, #362761) for volunteers at various time points (day 0, 21, 90 and 180 post first-dose and ~1 month post the booster dose, which was administered between days 189–270 post first-dose[144]).

### Serological assays for the detection of anti-SARS-CoV-2 antibodies
Serum specimens were stored at −25 °C and equilibrated at room temperature before time of analysis. Samples were analyzed using two commercial assays, in accordance with the manufacturer's protocol. The anti-SARS-CoV-2 S (Roche S) and anti-SARS-CoV-2 (Roche N) immunoassays using the Roche Cobas e411 Analyzer (Roche) allow the quantitative detection of total antibodies against the SARS-CoV-2 spike (S) protein receptor binding domain (RBD) and the qualitative detection of total antibodies against the SARS-CoV-2 nucleocapsid (N) antigen respectively. Plasma were incubated with either a mix of biotinylated and ruthenylated SARS-CoV-2 S-RBD antigens or N antigens to form immune complexes. Complexes were attached to streptavidin-coated microparticles upon incubation and then transferred to a measuring cell. For the Roche S assay, the electro-chemiluminescent signal representing the level of antibodies was measured and samples within the linear range of quantitation (0.4–250 U/mL) were assigned a value. Samples with antibody levels ≥0.8 U/mL were considered positive. For the Roche N assay, the cut-off index (COI) was derived from the measured signal, where samples with COI ≥ 1.0 were considered reactive.

### Spike protein flow cytometry-based assay (SFB assay) for antibody detection
Cells expressing the S-protein of the ancestral Wuhan strain, or of the Delta (B.1.617.2) or the Omicron (B1.1.529, BA.1 substrain) variants on the cell surfaces were used in this study[61,62]. Expression of the various S protein was verified with a serum from a vaccinated individual who recovered from a previous COVID-19 infection before vaccination (Supplementary Fig. 4). Cells were seeded at $1.5 \times 10^5$ cells per well in 96 well V-bottom plates. Cells were incubated with human serum (diluted 1:100 in 10% FBS) followed by a secondary incubation with a double stain, comprising Alexa Fluor 647-conjugated anti-human IgG (1:500 dilution) and propidium iodide (PI; 1:2500 dilution). Cells were acquired using a BD Biosciences LSR4 laser and analyzed using FlowJo (version 10, Tree Star). Gating strategies to determine spike-specific antibody response is described in Supplementary Fig. 5. The assay was performed as two independent experiments within technical duplicates each time.

### Pseudovirus neutralization assay
Pseudoviruses were produced as previously described[145,146] and the pseudotyped lentivirus neutralization assay was performed as previously described[147,148]. Briefly, a stable cell line expressing human ACE2, CHO-ACE2 (a kind gift from Professor Yee-Joo Tan, Department of Microbiology, NUS and IMCB, A*STAR, Singapore)[149] were used for the assay. CHO-ACE2 cells were seeded at $1.8 \times 10^4$ per well in a 96-well black microplate (Corning) in culture medium without Geneticin overnight. Serially diluted heat-inactivated plasma samples ($n = 29$ vaccine breakthrough, $n = 86$ close contact) at 1:5 to 1:5120 in four-fold serial dilutions were incubated with equal volume of pseudovirus expressing SARS-CoV-2 S proteins of either ancestral wildtype, Delta variant, or Omicron variant (5 ng p24 per well) at 37 °C for 1 h, before being added to pre-seeded CHO-ACE2 cells in duplicate. Cells were refreshed with culture media after 1 h incubation. After 48 h, cells were

washed with PBS and lysed with 1× Passive Lysis Buffer (Promega) with gentle shaking at 125 rpm for 30 min at 37 °C. Luciferase activity was subsequently quantified with Luciferase Assay System (Promega) on a GloMax Luminometer (Promega).

## Determination of SARS-CoV-2 neutralizing antibody level using sVNT

Neutralizing antibodies against SARS-CoV-2 was measured using the surrogate virus neutralization (sVNT) platform[41] and conducted according to manufacturer's protocol (cPass, GenScript). HRP-conjugated RBD (RBD-HRP) provided was diluted with HRP Dilution Buffer to 1:1000. Test plasma was diluted with the Sample Dilution Buffer to 1:10. The diluted plasma were then mixed with the diluted RBD-HRP in 1:1 ratio (e.g., 60 μL diluted plasma with 60 μL diluted RBD-HRP). The mixtures were incubated at 37 °C for 30 min. After first incubation, 100 μL of the mixtures was added into each well of the ACE2-coated plate provided. The plate was covered with a plate sealant and incubated at 37 °C for 15 min. After second incubation, the plate was washed four times with 260 μL 1× wash buffer to remove the unbound RBD-HRP. For measurement of RBD-HRP bound onto the plate, 100 μL of 3,3',5,5'-tetramethylbenzidine (TMB) was added into each well. The chromogenic signal was allowed to develop for 15 min in the dark before 50 μL of the TMB stop solution was added into the well. Absorbance at 450 nm was acquired using Cytation 5 microplate reader (BioTek). cPass percentage inhibition was calculated according to the manual, and a 30% cut-off was used to determine a positive result.

## Memory B cell ELISpot

SARS-CoV-2 RBD-specific memory B cell numbers were counted using the ELISpot Path: Human IgG (SARS-CoV-2, RBD) ALP kit (Mabtech), following manufacturer's instructions. Fresh PBMCs (1,000,000) were resuspended in 1 ml RPMI + 10% FBS + 1 μg/ml R848 + 10 ng/ml IL-2, and incubated at 37 °C, 5% $CO_2$ for 4–5 days to differentiate memory B cells into antibody-secreting cells. After incubation, cells were counted, and 100,000 or 400,000 live cells were taken for ELISpot plating to determine RBD-specific memory B cell numbers. Total IgG secreting cells were detected by plating 1500 or 3000 live cells to normalize the results. Plates were then read on an IRIS ELISpot reader (Mabtech). Spots were calculated based on the average of two wells using the MabTech IRIS Immunospot reader Apex software.

## Whole-blood culture with SARS-CoV-2 peptide pools

This was performed as described previously[69]. Whole blood (320 μl) drawn on the same day was mixed with 80 μl RPMI and stimulated with pools of spike protein peptides (2 μg/ml) (Supplementary Table 11) or a DMSO control. After 15 h of culture, the culture supernatant (plasma) was collected and stored at −80 °C until quantification of cytokines. Cytokine concentrations in the plasma were quantified using an Ella machine with microfluidic multiplex cartridges measuring IFN-γ and IL-2 following the manufacturer's instructions (ProteinSimple). The positivity threshold was set at 10 x times the lower limit of quantification of each cytokine (IFN-γ = 1.7 pg/ml; IL-2 = 5.4 pg/ml) after DMSO background subtraction.

## IFN-γ IL-2 FluoroSpot assays

Donor PBMCs were first thawed in RPMI-1640 with 10% Fetal Bovine Serum (R10 medium) and incubated overnight for recovery in high density (10 million PBMCs per 2 mL) in AB medium (RPMI-1640 + 10% Human AB Serum + 1% Penicillin Streptomycin + 1% 200 g/mL D-glucose). PBMCs were then used for FluoroSpot assays to measure CD8, CD4 Th1 and Th2 responses. CD8 and CD4 Th1 responses were measured using Human IFN-γ/IL-2 FluoroSpot PLUS kits as per manufacturer's protocol (Mabtech, Sweden). In brief, PVDF plates pre-coated with IFN-γ mAb (1-D1K) and IL-2 mAb (MT2A91/2C95) were

washed with sterile phosphate buffered saline (PBS) and blocked with R10 medium for at least 30 min at room temperature (RT). After overnight rest, PBMCs were harvested and suspended in AB medium. PBMCs were seeded at 250,000 cells per well and stimulated in duplicates with SARS-CoV-2 spike glycoprotein peptide pool (JPT Peptide Technologies, Germany) (Supplementary Tables 12 and 13) with 0.1 μg/mL co-stimulator anti-CD28 (mAb CD28A as per MabTech protocol). Medium containing 1% DMSO was used as negative control, while 0.02 μg/mL anti-CD3 mAb (CD3-2) was used as positive control. Cells were incubated overnight at 37 ˚C and 5% $CO_2$. Following overnight incubation, plates were washed with PBS and incubated with detection antibodies anti-IFN-γ mAb (7-B6-1-BAM) and anti-IL-2 mAb (MT8G10, biotinylated) diluted in PBS with 0.1% BSA for 2 h at RT. Plates were then washed with PBS and incubated with fluorophore conjugates for IFN-γ (anti-BAM-490) and IL-2 (SA-550) in PBS with 0.1% BSA for 1 h at RT. Plates were washed and incubated with ready-to-use fluorescent enhancer II for 15 min at RT. All incubations were performed in the dark. Plates were emptied and dried overnight at RT and analysed the next day with MabTech IRIS FluoroSpot and ELISpot reader using FITC filter (excitation 490 nm/emission 510 nm) for IFN-γ and Cy3 filter (excitation 550 nm/ emission 570 nm) for IL-2. Spots were calculated based on the average of two wells using the MabTech IRIS Immunospot reader Apex software.

## IL-4 IL-5 IL-13 FluoroSpot assays

CD4 Th2 responses were measured using custom Human IL-4/IL-5/IL-13 FluoroSpot FLEX kits as per manufacturer's protocol (MabTech). In brief, PVDF plates were activated with 15 μL 35% EtOH per well for a maximum of 1 min. Plates were washed with cell culture water and incubated with IL-4 mAb (IL4-I), IL-5 mAb (TRFK5) and IL-13 mAb (MT1318) in PBS at 4 °C overnight, protected from light. After overnight incubation, plates were washed with sterile PBS and blocked with R10 medium for at least 30 min at RT. After overnight rest, PBMCs were harvested and suspended in AB medium. Stimuli were prepared in AB media with 0.1 μg/mL co-stimulator anti-CD28 (mAb CD28A). PBMCs were seeded at 250,000 cells per well and stimulated with peptide pool from the Spike protein (JPT Peptide Technologies) (Supplementary Table 13). 1% DMSO only medium was used as negative control. Cells and stimuli were incubated overnight at 37 ˚C and 5% $CO_2$. Following overnight incubation, plates were washed with PBS and incubated with detection antibodies anti-IL4 mAb (IL4-II), anti-IL5 mAb (5A10) and anti-IL13 mAb (25K2) diluted in PBS with 0.1% BSA for 2 h at RT. Plates were then washed with PBS and incubated with fluorophore conjugates for IL-4 (SA-550), IL-5 (anti-WASP-640) and IL-13 (anti-BAM-490) in PBS with 0.1% BSA for 1 h at RT. Plates were washed and incubated with ready-to-use fluorescent enhancer II for 15 min at RT. All incubations were performed in the dark. Plates were emptied and dried overnight at RT and analysed the next day with Mabtech IRIS FluoroSpot and ELISpot reader using Cy3 filter (excitation 550 nm/ emission 570 nm) for IL-4, Cy5 filter (excitation 640 nm/ emission 660 nm) for IL-5 and FITC filter (excitation 490 nm/emission 510 nm) for IL-13. Spots were calculated based on the average of two wells using the MabTech IRIS Immunospot reader Apex software.

## Low responder population definition

Low responders were defined as fully vaccinated individuals with antibody response below cohort's median response at consecutive timepoints (%). As an example, the low responders, at day 90, had responses below cohort's median response at both day 21, 90 and 180.

## Statistical analysis

To assess immune positivity after the vaccine doses, we used two methods to define baseline values. In method 1, we define a cohort baseline using the upper range of the data set (for data which did not follow a normal distribution). For data which follow a normal

distribution, a cohort mean ± 3 SD was used to define a cut-off. In method 2, positivity was defined when values were above individual baseline values. Statistical analysis was performed using GraphPad Prism 9 software. Paired comparisons for samples taken at different times or unmatched pairwise comparisons (when comparing between age group) were performed using the Mann–Whitney U test, while matched pairwise comparisons were performed using the Wilcoxon matched pairs signed rank test. To compare between multiple groups, Kruskal–Wallis tests and post hoc tests using Dunn's multiple comparison tests were used to identify significant differences. Spearman's correlation analyses were performed to calculate correlation coefficient rho and $P$-value. $P$-values < 0.05 were considered significant. Statistical analysis was performed using GraphPad Prism 9.1.2. FluoroSpot results were analyzed with Welch's $t$-test for parametric unpaired comparisons. All tests were two-tailed and $p < 0.05$ was considered statistically significant.

## Reporting summary
Further information on research design is available in the Nature Research Reporting Summary linked to this article.

## Data availability
All other data are available in the article and its Supplementary files or from the corresponding author upon reasonable request. The source data for all figures are provided as a Source Data file. Raw data for the flow cytometry or ELISPOT can be obtained upon request to the corresponding author. Source data are provided with this paper.

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

## Acknowledgements

The authors would like to thank the study participants who donated their blood samples to this study. The authors also wish to thank the National Center for Infectious Diseases SCOPE team for their help in patient recruitment. This work was supported by the Biomedical Research Council (BMRC), A*CRUSE (Vaccine monitoring project), the A*ccelerate GAP-funded project (ACCL/19-GAP064-R20H-H) from Agency of Science, Technology and Research (A*STAR), Singapore National Medical Research Council COVID-19 Research Fund (COVID19RF-001; COV-ID19RF-007; COVID19RF-0008; COVID19RF-011, COVID19RF-060) (LR, LFPN, DCL, BEY), US Food and Drug Administration (#75F40120C00085) (LR, LN), and A*STAR COVID-19 Research funding (H/20/04/g1/006) (LR, WCI, LFPN).

## Author contributions

Conceptualization: L.R., D.C.L., Y.-S.L., R.T.P.L., L.-F.W., E.C.R., A.B., B.E.Y., and L.F.P.N. Sample collection: Y.D., S.P., L.J.S., J.S., Y.S.L., D.L.S.O., D.C.L., B.E.Y. Materials: P.A.M., C.I.W. Formal analysis: Y.S.G., A.R., N.L.B., W.N.C., J.-M.C., S.-W.F., L.R., D.C.L., Y.-S.L., R.T.P.L., L.-F.W., E.C.R., A.B., B.E.Y. and L.F.P.N. Investigation: Y.S.G., A.R., N.L.B., W.N.C., J.-M.C., S.-W.F., Z.W.C., N.Z.Z., M.Z.T., J.X.E.W., Y.-H.C., N.K.-W.Y., S.N.A., Y.H., P.X.H., C.Y.L., B.W., E.Z.X.N., S.N.M.S., G.C., S.D., A.J.L., C.W.T., J.Z., J.M.E.L. Writing—ancestral draft preparation: L.R., Y.S.G., D.C.L., A.B., E.C.R. Writing: review and editing: all authors. Supervision: L.R., L.F.P.N., D.C.L., B.E.Y., C.I.W., R.T.P.L., L.-F.W., E.C.R., A.B., P.A.M.

## Competing interests

A patent application for the SFB assay has been filed (Singapore patent #10202009679 P: A Method Of Detecting Antibodies And Related Products) (L.R., Y.S.G., and L.F.P.N). The authors declare no other competing interests.

## Additional information

[1]A*STAR Infectious Diseases Labs (A*STAR ID Labs), Agency for Science, Technology and Research (A*STAR), Singapore, Singapore. [2]Lee Kong Chian School of Medicine, Nanyang Technological University, Singapore, Singapore. [3]School of Biological Sciences, Nanyang Technological University, Singapore, Singapore. [4]Programme in Emerging Infectious Diseases, Duke-NUS Medical School, Singapore, Singapore. [5]National Public Health Laboratory, Singapore, Singapore. [6]National Centre for Infectious Diseases, Singapore, Singapore. [7]Singapore Immunology Network, Agency for Science, Technology and Research (A*STAR), Singapore, Singapore. [8]Ng Teng Fong General Hospital, Singapore, Singapore. [9]Alexandra Hospital, Singapore, Singapore. [10]Division of Infectious Diseases, Department of Medicine, National University Hospital, National University Health System, Singapore, Singapore. [11]National healthcare group polyclinic, Jurong, Singapore. [12]National University Polyclinics, National University of Singapore, Singapore, Singapore. [13]Saw Swee Hock School of Public Health, National University of Singapore, Singapore, Singapore. [14]Tan Tock Seng Hospital, Singapore, Singapore. [15]Department of Microbiology and Immunology, Yong Loo Lin School of Medicine, National University of Singapore, Singapore, Singapore. [16]Life Sciences Institute, Centre for Life Sciences, National University of Singapore, Singapore, Singapore. [17]Department of Medicine, Yong Loo Lin School of Medicine, National University of Singapore, Singapore, Singapore. [18]Department of Biochemistry, Yong Loo Lin School of Medicine, National University of Singapore, Singapore, Singapore. [19]National Institute of Health Research, Health Protection Research Unit in Emerging and Zoonotic Infections, University of Liverpool, Liverpool, UK. [20]Institute of Infection, Veterinary and Ecological Sciences, University of Liverpool, Liverpool, UK. [21]These authors contributed equally: Yun Shan Goh, Angeline Rouers, Nina Le Bert, Wan Ni Chia, Jean-Marc Chavatte, Zi Wei Chang, Nicole Ziyi Zhuo, Matthew Zirui. Tay. [22]These authors jointly supervised this work: Raymond Tzer Pin Lin, Lin-Fa Wang, Ee Chee Ren, David C. Lye, Antonio Bertoletti, Barnaby Edward Young, Lisa F. P. Ng. ✉e-mail: renia_laurent@idlabs.a-star.edu.sg

## SCOPE Cohort Study Group

**Anthony Torres Ruesta**[1], **Vanessa Neo**[1], **Wendy Yehui Chen**[1], **Estelle Yi Wei Goh**[1], **Alice Soh Meoy Ong**[1], **Adeline Chiew Yen Chua**[1], **Samantha Yee Teng Nguee**[1], **Yong Jie Tan**[1] & **Weiyi Tang**[1]

