## [Peer Review File · Nature Communications]

Lower vaccine-acquired immunity in the elderly population following two-dose BNT162b2 vaccination is alleviated by a third vaccine doseREVIEWER COMMENTS

Reviewer #1 (Remarks to the Author):

The results of humoral and T-cell immunity after double vaccination with the vaccine BNT162b2 in a cohort of 312 subjects are described and the efficacy with respect to the Delta Varinate and the Wuhan strain of SARS-CoV-2 was considered.

Already 95% of the examined subjects showed seroconversion after the first vaccination. Overall, the results of the determination of antibody levels at 21 and 90 days showed strong differences in various functional tests taking age into account. All vaccinees older than 60 years showed a significantly low response and a more pronounced decline compared to the younger ones. Regarding T cell responses determined in 155 randomly selected vaccinees by secretion of IL-2 and IFN-gamma, it was shown that 100% of the vaccinees had IL-2- and approximately 95% IFN-gamma-secreting T cells detectable against the spike protein. In contrast to the antibody response, there were no differences with respect to age in the CD8 T cell-mediated responses. CD4+ SARS-CoV2-S specific T cells were more frequent in the older cohort.

The data are solidly presented and the statistical analyses used are appropriate.

Points of criticism:

- Line 237 states that 155 individuals were selected; line 238 states 81 and 82 (= 163). Please explain the difference.
- Line 439: it is not clear what the composition of the S-peptide pool and the corresponding peptides is; multiple pools were described in Tan et al., 2021 (JCI)
- can the different T cell reactivity be described by the composition of the pools? It is known that longer peptides are more likely to be recognized by CD4 T cells? What is the influence of the HLA expression of the vaccinees?

The data are solidly presented and the statistical analyses used are appropriate.

Points of criticism:

- Line 237 states that 155 individuals were selected; line 238 states 81 and 82 (= 163). Please explain the difference.

- Line 439: it is not clear what the composition of the S-peptide pool and the corresponding peptides is; multiple pools were described in Tan et al., 2021 (JCI)

- can the different T cell reactivity be described by the composition of the pools? It is known that longer peptides are more likely to be recognized by CD4 T cells? What is the influence of the HLA expression of the vaccinees?

Reviewer #2 (Remarks to the Author):

Summary: In the current study, the authors looked at the antibody and t cells responses in a cohort of vaccinated individuals up to 6 months after immunisation with the mRNA vaccine BNT162b2. They observed differences between the under and over 60s in the magnitude of the response and the decline of antibodies to spike.

Major comments:

1. For higher impact of the study, given the pandemic has moved on and there is another circulating strain with greater immune escape:

a. Studies comparing binding against omicron are needed.

b. Likewise, looking at the impact of a 3rd boost of the vaccine in the same cohort is important.

2. The title needs to be changed – comparing T cell responses measured by stimulating the cells with antibody levels in serum is not a direct comparison. As one is a re-stimulation and the other direct activity.

3. Presentation of figures:

a. There are a lot of data points, which makes interpreting the means hard. In Figures 1B-D it would help to have a red line to represent mean over the top of the spaghetti plots

b. The interpretation of the PCA plots seems wrong, the over and under 60 year olds are overlapping – there is no separation.

c. 2B the mean are very hard to see – can you increase the size of the bar? (and other plots where there are multiple points)

d. 2C – I think fold change for all of these measures would be better than absolute change then it is comparable across the different assays.

e. 3B might help to have a mean/ median bar in the plots

f. 4A and 4B are showing the same thing – why present twice?

g. I don't agree with the CD8/ CD4 distinction based on elispot and peptide length alone. I think it is more accurate to say response to short peptide pool and long peptide pool as the cells haven't been sorted.

4. The introduction needs more on other studies that have looked at longitudinal vaccine responses.

Minor comments:

1. Not clear what a 'secondary' anti-RBD response is

2. Lines 190 and 297 seem to contradict each other – one says decline is greater, the other is that it is larger.

Response to reviewers

Reviewer #1 (Remarks to the Author):

The results of humoral and T-cell immunity after double vaccination with the vaccine BNT162b2 in a cohort of 312 subjects are described and the efficacy with respect to the Delta Variant and the Wuhan strain of SARS-CoV-2 was considered. Already 95% of the examined subjects showed seroconversion after the first vaccination. Overall, the results of the determination of antibody levels at 21 and 90 days showed strong differences in various functional tests taking age into account. All vaccinees older than 60 years showed a significantly low response and a more pronounced decline compared to the younger ones. Regarding T cell responses determined in 155 randomly selected vaccinees by secretion of IL-2 and IFN-gamma, it was shown that 100% of the vaccinees had IL-2- and approximately 95% IFN-gamma-secreting T cells detectable against the spike protein. In contrast to the antibody response, there were no differences with respect to age in the CD8 T cell-mediated responses. CD4+ SARS-CoV2-S specific T cells were more frequent in the older cohort.

The data are solidly presented, and the statistical analyses used are appropriate.

We thank the reviewer for her/ his positive comments.

Points of criticism:

- Line 237 states that 155 individuals were selected; line 238 states 81 and 82 (= 163). Please explain the difference.

We apologize for the mistake. This was corrected.

- Line 439: it is not clear what the composition of the S-peptide pool and the corresponding peptides is; multiple pools were described in Tan et al., 2021 (JCI)

We have added the peptides used for the whole blood peptide stimulation in Supplementary Table 11 and the peptides used for the ELISPOT in supplementary Tables 12 (CD4) and 13 (CD8).

- can the different T cell reactivity be described by the composition of the pools? It is known that longer peptides are more likely to be recognized by CD4 T cells? What is the influence of the HLA expression of the vaccinees?

We respectfully disagree with the reviewer here. There is an assumption that long peptide equate CD4 cells. However, we and others have shown that this assumption is not completely true. We and others have reported very good CD8 T cell response using 15-mers (Kiecker et al, Hum Immunol, 2004, 65, 523-536; Zhang et al, J Biol Chem, 2009, 284, 9184-9191; Strynh et al, Front Immunol, 2020). The assumption might be true when 20-mers are used. Indeed, in a recent article, these longer peptides were slightly better at activating CD4 than CD8 (Gao et al, Nat Med, 2022. Doi: 10.1038/s41591-022-01700-x). For better illustration, please see figure below taken from the in press manuscript.

Hence, we wish to clarify the below:

- 1) We did not use 20-mers peptides for the whole blood peptide stimulation in this study, which at recall stimulation for all T cell subsets.
- 2) 15-mers peptides were used in the CD4 ELISPOT.

With regards to the influence of HLA expression, using 9-10 mers perfectly matched to HLA might be better than longer peptides in some circumstances, but would also like to caution against this as this is not always the case (see Zhang et al, J Biol Chem, 2009, 284, 9184-9191).

Reviewer #2 (Remarks to the Author):

Summary: In the current study, the authors looked at the antibody and t cells responses in a cohort of vaccinated individuals up to 6 months after immunization with the mRNA vaccine BNT162b2. They observed differences between the under and over 60s in the magnitude of the response and the decline of antibodies to spike.

Major comments:

1. For higher impact of the study, given the pandemic has moved on and there is another circulating strain with greater immune escape:

We agree with the reviewer and are providing new data on the Omicron variant. We would like to highlight that we have submitted this article in November, and, at that time the Delta variant was prominent in Singapore (over 95% of the cases) and the Omicron variant has just been detected in Botswana and South Africa.

a. Studies comparing binding against omicron are needed.

We agree with the reviewer and have provided data on the Omicron variant. See revised Figures 3A, 7B, and 7E.

b. Likewise, looking at the impact of a 3rd boost of the vaccine in the same cohort is important.

We agree with the reviewer that looking at the impact of a 3rd boost is important. However, due to the government decision, the elderly and, in particular, the low responders were prioritized for booster vaccination. Thus, considering the time frame for the revision of this article, we only have data on booster vaccination in this category of individuals, with sufficient n number. We have added these data as a new figure (Figure 7).

2. The title needs to be changed – comparing T cell responses measured by stimulating the cells with antibody levels in serum is not a direct comparison. As one is a re-stimulation and the other direct activity.

We have now amended the title, in consideration of the reviewer's comment and the new set of data provided.

3. Presentation of figures:

a. There are a lot of data points, which makes interpreting the means hard. In Figures 1B-D it would help to have a red line to represent mean over the top of the spaghetti plots.

As requested by the reviewer, we have added a red line representing the median over the top of the plot. The median was chosen over the mean because the data distribution is not normal.

b. The interpretation of the PCA plots seems wrong, the over and under 60-year-old are overlapping – there is no separation.

We do agree that the separation is less evident in the PCA plots. Thus, we have removed this figure

c. 2B the mean are very hard to see – can you increase the size of the bar? (and other plots where there are multiple points)

The size of the bar and the plots has been increased

d. 2C – I think fold change for all of these measures would be better than absolute change then it is comparable across the different assays.

We disagree with the reviewer. We have two assays, one expressed as percentages and the other in arithmetic numbers. The fold changes thus will not be directly comparable.

e. 3B might help to have a mean/ median bar in the plots

As requested by the reviewer, we have added a red line representing the median over the top in the plot.

f. 4A and 4B are showing the same thing – why present twice?

We would like to clarify that Figure 4A and 4B do not represent the same thing. In Figure 4A, we have a larger number of individuals, but there are fewer points at some time points (due to insufficient numbers of cells to perform the assay). In Figure 4B, we have showed paired samples for which we have all time points. This was indicated in the figure legend.

g. I don't agree with the CD8/ CD4 distinction based on elispot and peptide length alone. I think it is more accurate to say response to short peptide pool and long peptide pool as the cells haven't been sorted.

We have discussed this point above and have amended the text to take into account both reviewer #1 and #2 comments.

4. The introduction needs more on other studies that have looked at longitudinal vaccine responses.

We have added additional references corresponding to the recent studies on longitudinal vaccine responses.

Minor comments:

1. Not clear what a 'secondary' anti-RBD response is

The sentence has now been amended to make it clearer.

2. Lines 190 and 297 seem to contradict each other – one says decline is greater, the other is that it is larger.

This part has now been amended for better clarity.

REVIEWERS' COMMENTS

Reviewer #1 (Remarks to the Author):

The concerns raised during the first revision were addressed by the authors. However, for one of the points of criticism, regarding the inconsistency in n number (n=155 vs. n=82 & n=75; line 224), the correction is not recognizable from the revised manuscript. Please revise.

Reviewer #2 (Remarks to the Author):

thanks for making the changes and including the new data.

Reviewer #1 (Remarks to the Author):

The concerns raised during the first revision were addressed by the authors. However, for one of the points of criticism, regarding the inconsistency in n number (n=155 vs. n=82 & n=75; line 224), the correction is not recognizable from the revised manuscript. Please revise.

We apologize for the omission. This was corrected in the main text, line 220 (This was done in a subset of volunteers (n=160) randomly selected from the cohort but age-matched (n=82 < 60 and n=78 ≥ 60). and in the Figure 5 legend , n=160).